# Characterization of Spontaneous Melanization by Fluorescence Spectroscopy: A Basis for Analytical Application to Biological Substrates

**DOI:** 10.3390/biology12030433

**Published:** 2023-03-11

**Authors:** Anna Cleta Croce, Francesca Scolari

**Affiliations:** 1Institute of Molecular Genetics, Italian National Research Council (IGM-CNR), Via Abbiategrasso 207, I-27100 Pavia, Italy; 2Department of Biology & Biotechnology, University of Pavia, Via Ferrata 9, I-27100 Pavia, Italy

**Keywords:** emission spectra, excitation spectra, melanin, L-tyrosine, Asian tiger mosquito, *Aedes albopictus* eggs, black cat hairs

## Abstract

**Simple Summary:**

Melanin is a pigment naturally present in various living beings, from animals to plants and fungi. Melanin can be involved in many processes, such as innate immunity and antioxidant protection, including photoprotection and the related abilities to confer color to many biological structures and to interact with light. Light interaction, in particular, is largely exploited to investigate melanin chemical components and precursors for various purposes, from the in situ grading of the pigment for diagnosis in biomedicine to the management of light energy for the development of innovative applications in bioengineering. In this study, the ability of melanin and its precursors to respond to light irradiation have been characterized in terms of changes in the emission light signals, namely fluorescence. Besides the confirmation of the fluorescence emission signature of melanin and of its precursor L-tyrosine, new insights have been provided on intermediates during melanin production and by different conditions of measurement. The subsequent characterization of aqueous extracts from black cat hairs and mosquito eggs proved the value of these experimental procedures for application in the investigation of melanization processes in biological substrates.

**Abstract:**

Melanin is present in various biological substrates where it may participate in several processes, from innate immunity to the still-unsolved opposite roles in antioxidant protection, including photoprotection and the related ability to interact with light. Melanin–light interaction has also been an important source of inspiration for the development of innovative bioengineering applications. These are based on melanin’s light-energy-absorption ability of its chemically and structurally complex components and precursors, and on the improvement in analytical and diagnostic procedures in biomedicine. In this regard, here, we characterized the fluorescence spectral properties of melanin and of its precursor L-tyrosine in an aqueous solution during spontaneous melanization. Besides the confirmation of the typical fluorescence-emission signature of melanin and L-tyrosine, we provide additional insights on both emission and excitation spectra recorded during melanization. On these bases, we performed a subsequent characterization on the aqueous extracts from two different melanin-containing biological substrates, namely hairs from a domestic black cat and eggs from the Asian tiger mosquito. The results from the mild extraction procedure, purposely applied to obtain only the soluble components, combined with fluorescence spectral analysis are expected to promote further investigation of the melanization processes, particularly in insects.

## 1. Introduction

Melanin is a natural biopolymer ubiquitously distributed in most living beings, from fungi to different taxa of plants and animals [1,2,3]. Abundant amounts of melanin are usually obtained from the common cuttlefish, *Sepia officinalis*, to be used for multiple purposes, from the study of its composition and biophysical properties to pharmaceutical and industrial applications or for alimentary use [4,5]. The generally known ability of many cephalopods to produce and eject melanin in their natural environment as an ink for defense and interaction between prey and predator is a primary example of the various biological functions played by melanin and its precursors, also considering that melanin shares similar intermediate products with biomolecules acting as hormone regulators [6]. The photoprotection and tanning of human skin are within the generally recognized effects of melanin [7]. In addition, hypotheses about the contrasting roles of melanization processes in the development of skin cancer and melanomas, as well as in the protection from such diseases, have been formulated [6,8,9]. In higher animals, neuromelanin produced from the oxidation of dopamine and other catecholamines in dopaminergic neurons accumulates in defined sites of the central nervous system, such as in the *substantia nigra* and in the *locus coeruleus*, where positive or negative outcomes may depend respectively on its ability to bind metal ions and toxic compounds or to leak from dying neurons and activate microglia, inflammation and neurodegeneration [10,11]. Melanin can also act as antimicrobial and antiparasitic agent, contributing to the innate immune response in the skin and various tissues of humans, as well as in different animals including insects [12,13,14,15,16].

The presence of melanin and its precursors in different body sites in an organism is thus relevant for its various, multifaceted, interlaced and still not completely characterized biological roles, which is, in turn, relatable to its biochemical, photophysical and antioxidant properties [1,2,3,17,18].

The production of melanin from tyrosine occurs through oxidative reactions catalyzed by the enzyme tyrosinase or spontaneously in the presence of oxygen, with the subsequent aggregation of oligomeric intermediates. Among the possible various intermediates, 5,6-dihydroxyindole and 5,6-dihydroxyindole-2-carboxylic acids have been identified as the precursors able to combine in macromolecules with a complex and disordered secondary structure in dark brown eumelanin. The involvement of cysteine for the synthesis of cysteinyl-DOPA is distinctive in the production of the reddish pheomelanin, less photostable and more pro-oxidant than eumelanin [6,18,19].

The melanization process entails the production of many chemically heterogenous intermediates. Analytical procedures like HPLC, electrospray ionization mass-spectrometry, matrix-assisted laser desorption/ionization mass-spectrometry or pyrolysis gas chromatography, as well as ultrastructural imaging techniques, have been applied to detect specific oxidative or hydrolytic degradation products of melanin to study synthetic melanogenesis and its regulation in cells, or to investigate the structural organization of natural melanin [19,20,21,22,23,24,25,26]. The comparison of data from various studies that has been proposed is hardly able to provide equivalent analytical results, which can provide complementary information useful to characterize melanin of various biological substrates for different applications [26]. In any case, the above-mentioned analytical procedures are complex, costly, time-consuming and limited to the defined time points of the sample collection. Conversely, optical spectroscopy and imaging, although unable to specifically detect single intermediate species, are suitable to perform real-time, in situ, repeatable assays to monitor melanization and detect melanin.

The well-assessed melanin light-absorption properties consist in a continuous curve which decreases monotonically from near-UV along the visible spectrum. As for fluorescence, the increase in the excitation wavelengths results in melanin emission curves sharing their boundary at their longest wavelength side, with peaks localized at progressively longer wavelengths and with lower amplitudes [27,28,29].

Contrarily to melanin, its chemical precursor L-tyrosine exhibits well-defined and narrower absorption and fluorescence excitation and emission bands. The optical properties of L-tyrosine have been initially characterized along with those of phenylalanine and tryptophan in the near-UV–blue spectral region, to provide a basis for further studies on peptides and proteins [30,31,32]. These three aromatic amino acids share similar absorption/excitation optical profiles with two bands, the major one peaking around 220 nm, and a minor and broader one peaking at about 260 nm, 275 nm and 280 nm for phenylalanine, L-tyrosine and L-tryptophan, respectively. Emission spectra, in turn, respectively peak around 280 nm, 300 nm and 350 nm, indicating that the different substituents of these aromatic amino acids affect the absorption/excitation spectral profiles less than the fluorescence emission properties [30]. A red emission band above 600 nm for both L-tyrosine and L-tryptophan has been also reported [33].

Up to now, absorption spectroscopy has been the foremost optical technique used to follow the production and aggregation of intermediates during the melanization process starting from the oxidation of tyrosine and dopamine, as well as the presence of specific components. Different experimental and computing studies based on spectroscopic simulation have shown the progressive increase in and broadening of the absorption bands in the 270–350 nm and 400–600 nm spectral ranges [33,34,35,36,37,38]. Fluorescence investigation has considered only some specific indole derivatives as precursors or components of eumelanin, with their possibly related emission and excitation bands [29,39,40].

Different kinds of diagnostic and analytical procedures are increasingly considering the optical properties of melanin as a useful intrinsic biomarker. In histopathology, melanin can hinder the bright-field-microscope observation of morphology and expression markers of stained and immune-stained cells and tissues. This problem can be solved with a suitable light irradiation of the tissue slice to induce photobleaching and the consequent disappearance of the dark pigment, as verified for melanoma samples from different veterinary sources [41]. Conversely, sophisticated techniques such as multiphoton excitation and lifetime spectroscopy and imaging focus on melanin as an optical parameter for the real-time, in vivo screening of skin disease and diagnosis of melanoma [42,43,44,45,46,47]. Similarly, in ophthalmology, melanin is characterized along with lipofuscins for the in vivo optical detection and monitoring of retinal aging and diseased conditions [48,49,50].

In insects, the optical properties of melanin are widely considered for studying the melanization processes, considering its importance in immunity, as well as the role of pigment contribution to the body biocomponents relevant to the hardening of the superficial cuticle and pigmentation patterns, with related behavioral implications [3,14,51,52,53,54]. In a recent study, melanins have been suggested to contribute to the autofluorescence of the reddish components of some brownish scales covering the body of the Asian tiger mosquito, *Aedes albopictus* (Skuse, 1894) (Diptera, Culicidae) [55]. These findings opened novel opportunities to the investigation of mosquito optical properties, a recently expanding research field [55,56,57,58], either to develop new tools for vector tracing or to shed light on potential fluorescence-based communication strategies in mosquitoes.

In this context, the aim of this work is to set up a simplified, optically based procedure to help the investigation of the melanization processes in different biological substrates. To this end, spectroscopy was applied to assess fluorescence excitation and emission changes in L-tyrosine undergoing spontaneous oxidation in an aqueous solution, up to the formation of synthetic eumelanin. The experimental excitation and emission spectra of L-tyrosine, L-tryptophan and melanin were recorded and are shown in combination as reference data to allow for an easier and more direct appreciation of the spectral shape changes occurring during melanization.

Throughout this study, eumelanin will be named “melanin”. L-tryptophan was also considered, because its presence in biological substrates may contribute to their fluorescence. The spectroscopic results from these synthetic compounds were then used for a comparative investigation on the presence of melanin intermediates in aqueous solutions obtained by a mild extraction procedure from the hairs of a black domestic cat and the eggs of *Ae. albopictus*, as natural sources of melanin in phylogenetically distant animal species [59,60,61].

## 2. Materials and Methods

### 2.1. Chemicals

L-tyrosine and L-tryptophan were obtained as dry powders (both BioUltra-≥99.0%-degree, Sigma Chem Co. St Louis, MO, USA) to be dissolved in phosphate buffer (pH 7.2, 0.1 M) as saturated stock solutions (5.0 mg/mL and 11 mg/mL, respectively). The solutions were preserved in the dark, in humidified air to allow exposure to atmospheric oxygen and consequent oxidation of the amino acids. A 24-month-old solution provided synthetic melanin, as a completed melanization product.

### 2.2. Biological Samples

Black hairs were collected by brushing a domestic black cat (*Felis catus* Linnaeus, 1758) with a soft brush. Mosquito eggs from the *Ae. albopictus* RER strain [62], which was obtained from the Centro Agricoltura Ambiente (CAA) “G. Nicoli” (Crevalcore, Italy) in 2022, were collected (about n = 30 eggs) and immediately processed for the aqueous extract preparation. The biological samples were rinsed three times with ultrapure water before the further processing with the gentle mild procedures, purposely applied to preserve the chemical and structural properties of the investigated biomolecules.

Fresh biological samples were transferred to an isotonic solution (NaCl 0.9%; 2 mL) to be homogenized using an immersion homogenizer (Ultra-Turrax T25, IKA Werker, Janke & Kunkel, Breisgau, Germany). The solutions were centrifuged, and the supernatant was collected to be preserved at −20 °C until they were subjected to spectrofluorometric analysis. For each sample, the procedure lasted a total of two minutes. At the end of each preparation, the homogenizer-dispersing tool was removed and cleaned following three cycles of sonication (5 min each) and rinsing with ultrapure water.

Pictures of both mosquito eggs and black cat hairs were collected using a stereomicroscope (Olympus SZ40, Olympus Optical Co. GmbH, Hamburg, Germany), coupled with a smartphone camera.

### 2.3. Spectrofluorometric Analysis

The fluorescence excitation and emission spectra were recorded from the L-tyrosine and L-tryptophan solutions immediately after their preparation and at different times from the L-tyrosine solution preparation, or from the biological sample solutions by means of the LS 55 Model spectrofluorometer (PerkinElmer Italia, Milan, Italy). All measurements were performed at room temperature. The solutions from the pure compounds were subjected to consecutive sets of measurements by varying the excitation and emission conditions, to detect the different excitation and emission bands and define the related measurement conditions to be applied for the subsequent spectrofluorometric study of the biological sample preparations. To minimize inner-filter effects, the optical density at the spectral positions of interest was checked by a spectrophotometer (Spectronic Genesys Spectrophotometer, Spectronic Instruments Inc., Rochester, NY, USA), and proper dilutions were prepared at the convenience of the linearity of the fluorescence response.

The emission bands detected from the biological sample solutions by exciting at 220 nm were analyzed by means of a curve-fitting procedure, to validate and estimate the effective contribution of L-tyrosine and L-tryptophan. The Marquardt–Levenberg algorithm [63] was applied using the PeakFit program (Peak fit; SPSS Science, Chicago, IL, USA) first to define the parameters representing the emission bands of L-tyrosine and L-tryptophan in terms of respective Half-Gaussian Modified Gaussian (GMG) spectral functions. The GMG functions were then used to analyze the emission of the biological samples, to detect and estimate the relative contribution of each amino-acid emission to the overall band area. More details on the procedure are provided in the Appendix A.

## 3. Results

### Synthetic Melanin and Pure Compounds

Synthetic melanin in solution, excited at wavelengths ranging from 230 to 500 nm, shows a series of emission curves. The increase in the excitation wavelength results in a shift in the peak positions from about 440 nm to 560 nm, and in a parallel decrease in the signal amplitudes.

Only the shortest excitations at 230 nm and 270 nm result in the presence of a minor band centered at about 360 nm (Figure 1A). Excitation profiles consist in a band centered at about 280 nm, when recorded at 320 nm and 360 nm. Observations at longer wavelengths show signal profiles decreasing from 200 nm to 290 nm, as well as signal profiles covering increasingly longer spectral ranges depending on the lengthening of the recording wavelength.

A band covering the range between 295 nm and 400–420 nm is observed at both 420 nm and 430 nm. An additional band in the 370–500 nm range is observed at 510 nm and 530 nm, and a lower and less structured emission profile is recorded at 560 nm. Observations at longer wavelengths, from 600 nm to 640 nm, show a very low signal profile in the 220–300 nm range, followed by an almost negligible emission at longer wavelengths (Figure 1B).

Following the settlement of the optical features of synthetic melanin in solution, the fluorescence excitation and emission properties of L-tyrosine were recorded to provide initial reference measurement conditions for the subsequent study of optical changes occurring during melanization (Figure 2A).

Emissions excited at both 230 nm and 270 nm consist in a band in the 280–350 nm range, peaking at about 310 nm. An additional band is detected in the red region, with the maximum around 610 nm. Excitation, in turn, consists in a signal decreasing rapidly from 220 nm to a minimum at 240 nm, followed by a band centered at about 270 nm. This profile is observed at both 310 nm and 610 nm, while negligible fluorescence signals are obtained under different excitation and emission conditions (Figure 2A). In comparison with the spectra of L-tyrosine, L-tryptophan shows similar fluorescence-excitation profiles, while emission spectra recorded in both the blue and red regions are wider and red-shifted, with the respective maximum values at about 360 nm and 860 nm (Figure 2B).

The fluorescence spectra recorded from L-tyrosine solution at one month of aging show the typical bands observed from the freshly prepared solution, accompanied by some changes in their profile and by the appearance of new spectral profiles. In more detail, excitation at 220 nm and 270 nm still results in the typical and well-defined emission bands centered at about 310 nm. Excitations longer than 300 nm result in a series of emission curves in the 350–600 nm spectral range, which in general undergo a decrease in emission amplitude depending on the lengthening of the excitation wavelength. In addition, excitation from 310 nm to 340 nm results in emission bands peaking at about 420 nm, while excitation at longer wavelengths results in an emission profile widening up to about 590 nm (Figure 3A).

Excitation profiles recorded at 310 nm, 610 nm and 640 nm are comparable to those detected from L-tyrosine, while additional bands appear in the 320–520 nm range when the signal is detected at longer wavelengths. In particular, a well-defined band peaking at about 320 nm is observed at 420 nm. This band shows a decrease in its amplitude when the recording wavelength is lengthened up to 530 nm, as well as the appearance of shoulders centered at about 380 nm and 445 nm (Figure 3B).

The fluorescence spectra recorded from the L-tyrosine solution at six months of aging still show the typical bands of L-tyrosine, in the UV- and red-emission regions. The curves detected in the 350–600 nm spectral range under increasing excitation wavelengths show a constant red-shift in their maximum peaks, in parallel with a decrease in their amplitude. Notably, these emission bands share a common boundary position at the longest wavelength, similarly to what is observed for the synthetic melanin (Figure 3C). Excitation spectral profiles are similar to those described for the one-month-aged L-tyrosine, except for the increase in the amplitude of the bands peaking at 380 nm and 445 nm when observed in the 420–530 nm range (Figure 3D).

Fluorescence spectroscopy revealed a series of emission and excitation bands with different peak positions and spectral profiles, in a close dependence on the changes in the respective excitation and observation wavelengths. A sensitive and more variegated response is thus provided by fluorescence analysis, compared with the smooth spectral profiles of the absorption spectra, as shown by our direct measurements (Appendix A) and by the literature [28,38].

The spectroscopic analysis of the pure compound solutions was followed by the characterization of the aqueous extract preparations obtained from the two different biological samples, namely the black cat hairs (Figure 4A) and the eggs of *Ae. albopictus* (Figure 4B).

Fluorescence spectra recorded from the solution prepared from the black cat hairs show three well-defined emission bands peaking at about 340 nm, and two bands in the 600–720 nm interval, when excited in the 220–290 nm range. Much lower emission profiles, with maxima at about 420 nm, are observed under excitation at wavelengths longer than 310 nm (Figure 5A).

Excitation spectra observed at both 310 nm and 340 nm or in the red region show a signal decreasing sharply from 220 nm to the minimum at about 240 nm, and a well-defined band peaking at about 270 nm. The excitation profiles observed in the 420–530 nm range show maximum peaks at about 320 nm, followed by a slow signal decrease up to 420 nm (Figure 5B).

The aqueous extract solution obtained from the mosquito eggs shows spectra similar to those measured from black cat hairs in the shorter spectral region, or in the red one. Much more defined excitation and emission profiles are detected in the 320–420 nm or in the 400–600 nm range, respectively. Interestingly, excitation at wavelengths longer than 310 nm results in a series of emission spectra showing a red shift in the maximum peak accompanied by a decrease in the amplitude, similarly to what is observed for both synthetic melanin and for the advanced melanization step (Figure 6A).

Excitation spectra exhibit a shift in the maximum values from about 340 nm to 360 nm depending on the lengthening of the observation, i.e., from 420 to 530 nm (Figure 6B).

Lastly, attention was given to the main emission bands of the biological samples excited at 220 nm, which are slightly red-shifted and wider compared to the emission of L-tyrosine, consistent with a possible contribution of L-tryptophan. These emission bands were thus subjected to a spectral fitting analysis procedure, which confirmed that both amino acids participate in the emission spectra excited at 220 nm (Appendix A). The relative areas covered by the signals ascribed to the two amino acids were also estimated, indicating that L-tryptophan contributes to the overall signal for a fraction slightly greater than 50% in both biological samples, while the relative contribution of L-tyrosine is about 28% in mosquito eggs and 13% in black cat hair solution extracts.

## 4. Discussion

The fluorescence emission spectra collected from L-tyrosine in freshly prepared solutions are fully consistent with their specific optical signatures described in the literature [30,31,33,64]. Additionally, synthetic melanin shows its typical continuous series of fluorescence-emission curves, characterized by the peaks undergoing a red shift and the amplitude lowering at increasing excitation wavelengths while sharing the same boundary at their red side. This is in agreement with the literature results on melanin fluorescence excited at wavelengths from 360 nm to 400 nm [27,28,29]. These optical results are commonly considered to be in keeping with the exponential decrease in the melanin-absorption-curve profile, interpreted in terms of a linear combination of successive, partially superimposed and individually unresolvable Gaussian curves. These curves, in turn, have been proposed to reflect comparable, although chemically different, species, with different polymerization degrees accounting for a progressive weakening of the strength of energy transitions at increasing measurement wavelengths [65]. In this respect, it is also worth recalling that, similarly to humic acids, the continuous and broad absorption spectrum of melanin could result from a system of continuous electronic levels, favorable for electronic excitation and transitions relevant to the induction of the “chemiluminescence” events activated by the photo-oxidation of synthetic melanin [66]. Subsequent works have substantiated the concept of the common system of electronic levels resulting from the disordered network of aromatic species, able to favor the energy transitions matching with the absorption and emission of light in the visible range [40,65]. A more recent work based on ultrafast laser spectroscopy on the induction of excitons and charge transfer in the heterogenous chromophores of melanin have even compared the energetic behavior of melanin to that of a disordered graphene-like material [67]. In this respect, it is also worth recalling that the manipulation of the self-assembly process with changes in ions and pH in the production of liquid-like melanin have been explored, posing promising perspectives for innovative applications in bioengineering [68]. All the above recalled studies thus substantiate the extreme complexity of melanin, which does not affect its typical, well-assessed spectral signatures.

As for the excitation spectra, our synthetic melanin showed a sequence of excitation spectra which widen at their red side, depending on the lengthening of the observation wavelength. This finding is consistent with the few reports provided in the literature on melanin excitation spectroscopy [45]. In addition, the changes in the excitation profile, with the signal decrease from 220 nm and the subsequent appearance of two bands at longer spectral positions when lengthening the observation wavelength, recall the behavior shown by Teuchner and colleagues, who also verified that melanin fluorescence depended on solvent medium and on single- or two-photon excitation [45].

The initial oxidation of L-tyrosine and the subsequent production and aggregation of intermediates during the melanization process result in progressive changes in its optical properties. Different absorption-based studies have provided a general indication on the increase and widening of the bands in the 270–350 nm and 400–600 nm spectral ranges detected after the irradiation or oxidation of L-tyrosine and DOPA derivatives [33,36,37,38]. Comparable data have been computed by means of spectroscopic simulation based on the structure and energy status of species such as indole quinone reduced forms and their oligomers [34,35,69]. As for fluorescence, information is limited to few, specific indole-based intermediates. For example, the steady-state fluorescence emission spectrum of 5,6-dihydroxyindole has been found to exhibit a main band peaking at about 340 nm, as well as a band in the 500–600 nm spectral range detected from time-resolved spectra [39]. A synthetic N-methyl,5-hydroxy,6-methoxyindole homopolymer exhibited two wide-emission bands, one peaking at a wavelength slightly shorter than 500 nm when excited at 355 nm, and one peaking at slightly over 500 nm, independently from the increase in the excitation wavelength from 370 to 400 nm [40]. The collection of excitation spectra with a fixed peak at around 365 nm independent from the increase in the reading wavelengths suggests the presence of dihydroxyindole-carboxylic acid in the monomeric state or slightly bound to a synthetic eumelanin polymer [29].

Our data collected at intermediate stages of melanization have shown the appearance of new, well-defined emission bands in the 350–600 nm spectral range, rather than a widening of the still-present bands typical of L-tyrosine. In particular, the 366 nm excitation results in a wide emission covering the 420–520 nm range, while both 400 nm and 440 nm excitations result in two bands peaking at about 520 nm, similarly to what is reported in the literature for the N-Me-5H6MI homopolymer [40]. Excitation spectra, in turn, cover a 290–520 nm range, nearly comparable to what is reported in the literature for eumelanin [29]. In our case, anyway, observation at 420 nm reveals a 320 nm excitation band. The amplitude of this band decreases upon the lengthening of the measuring wavelength, accompanied by the rising of a shoulder at about 380 nm and of secondary band centered at about 440 nm. The changes in both excitation and emission profiles become more evident at longer time periods, after the preparation of the L-tyrosine solution, making them more similar to what is detected from the synthetic melanin solution. The typical bands of L-tyrosine, however, are hardly detectable in the synthetic melanin solution, indicating its almost complete transformation.

The above-described fluorescence data from the pure compound preparations, along with the available literature on the presence of melanin in biological samples [60,70], have been used to interpret the spectra from the aqueous extract solutions prepared from mosquito eggs and black cat hairs. The solution obtained from the mosquito eggs excited at wavelengths ranging from 310 to 440 nm showed well-defined emission spectra in the 350–600 nm range, in parallel with excitation bands in the 320–420 nm interval observed at wavelengths varying from 420 nm to 530 nm. All these results are in keeping with the presence of melanin compounds in the mosquito eggs, as previously shown in *Ae. albopictus* [70]. The preparation from black cat hairs, in turn, shows much lower signals in the spectral region typical of melanin compounds. The maxima of all emission spectra occur at about 420 nm, and the maxima of excitation spectra at about 320 nm. Excitation spectra also show a rapid decrease in the signal at their red side, independently from the lengthening of the observation wavelength, and can be easily distinguished from those relevant to L-tyrosine.

The comparison of the fluorescence results from the extract solutions prepared from the two different biological samples has revealed a remarkable difference as to the spectral region relating to melanin components. In this respect, it worth recalling that all our measurements were performed on aqueous solution extracts, similarly to the pure-compound preparations. The relatively gentle method adopted for the sample extraction was purposely chosen to obtain the most easily solubilizing compounds, preserving their chemical properties as much as possible compared to more aggressive treatments used for melanin isolation. In fact, these harsh methods can involve severe chemical treatments, such as extensive hydrolysis with boiling mineral acids or bases and followed by organic solvents’ washing to isolate melanin formed inside melanosomes, which are bound to cellular components such as minerals and proteins, and may result in alterations in the chemical and structural properties of melanin [24,26,61]. On these bases, the melanin tightly bound to the proteinaceous components of the hair’s mature melanosomes can likely hinder its extraction by our mild isolation method, thus accounting for our autofluorescence results on melanin extracts from black cat hairs [61,71].

Apart from the data of the spectral region relevant to melanin, both biological samples show the main emission band recorded by exciting at 220 nm, which is slightly red-shifted and wider than the emission of L-tyrosine. This finding indicates a contribution of L-tryptophan, which is typically fluorescent at a wavelength longer than that of L-tyrosine. The spectral fitting analysis validated the participation of the two amino acids in the overall emission of the 220 nm band, and also allowed us to estimate their relative contribution. L-tryptophan was indicated to provide a contribution higher than that of L-tyrosine in both biological samples, while the L-tyrosine signal is more than doubled in mosquito eggs compared to the cat black hairs. Two additional spectral components were required to achieve the goodness-of-fit of the procedure. The band peaking at about 310–320 nm is undefined since at the moment, there are no reference data available, while the peaking at about 410–420 nm can be related in general to a proteinaceous material [72].

Interestingly, the UV spectral component relatable to L-tyrosine and L-tryptophan is detected in parallel with the deep red emission still ascribable to these amino acids. For L-tyrosine in particular, this double signature, detected along with the spectra relatable to melanin components, is promising to estimate the degree of melanization process. In this regard, it is worth recalling the reports on the deep red emission detection in various melanin-containing samples of biomedical interest, in particular from skin and hairs, normal nevi and melanomas at different degrees of malignancy [42,43,45,46,47]. These findings could be accounted for by the influence of the environment and of the excitation technique, namely single-, two- or multiphoton excitation, on the fluorescence of melanin, although the presence of amino acids reflecting the tissue engagement in melanization cannot be disregarded.

In insects, melanin has been suggested to affect desiccation resistance through multiple mechanisms, depending on its covalent or noncovalent interaction with proteins, chitin and other biomolecules [73], or more generally on its hydrophobicity, which hampers the water flux through the cuticle [74]. In the eggs, melanin can decrease their porosity. This effect is particularly relevant for *Ae. albopictus*, whose eggs become dark black in a short time after oviposition and display high desiccation resistance, with a key impact on the geographical dispersal of this global mosquito invader [59]. Consistently, some of the commonly indicated yellow proteins have been demonstrated to participate in the enzyme conversion of dopachrome along the tyrosine-dependent pathway of tanning, favoring the rapid pigmentation and sclerotization of the chorion of *Ae. albopictus* eggs [70,75]. Implications on mosquito reproduction deriving from melanization effects on the cuticle mechanical stiffness may also regard some structures of adult insects. Indeed, the reduced melanization of the male-specific sex combs on the legs of *Drosophila melanogaster* has been shown to affect mating through changes in their structure, resulting in a diminished ability to grasp females before copulation [76].

More generally, it is also worth recalling that components of the melanin biosynthetic pathway have been previously shown to contribute to the wing-scale color of butterflies [51,77], and that melanization is known to play key roles in the structure and color of the cuticle [78,79,80].

## 5. Conclusions

Besides confirming the typical fluorescence spectral properties of L-tyrosine and melanin, variable excitation and emission spectral profiles were collected during the melanization process, providing a basis for the subsequent analysis of samples from black cat hairs and mosquito eggs. The use of these biological samples validated the ability of the mild extraction procedure applied to produce aqueous extracts suitable for the fluorescence analysis. The fluorescence spectra, in turn, were, in general, consistent with those observed from the pure compounds and during the melanization process. The lower signals from the black cat hairs, in turn, could be ascribed to the tight binding of melanin to the proteinaceous components of the hair’s mature melanosomes, hindering its extraction compared with mosquito eggs. Improvements in the fluorescence characterization of the melanization process in mosquitoes are expected to contribute to better clarify the coloring of biometabolic pathways, along with the roles of melanin and of its precursors relevant to eggs and cuticle functionality, as well as potential novel functions of fluorescence in mediating intra- and intersexual interactions. This additional optically based information in mosquito species may stimulate the development of novel approaches for tracing and controlling mosquitoes, as well as inspiring novel investigations on the evolutionary meaning of the coloration and fluorescence of eggs and adults.

## Figures and Tables

**Figure 1 biology-12-00433-f001:**
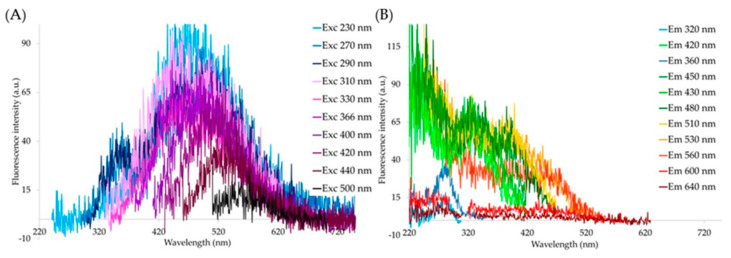
Emission (**A**) and excitation (**B**) spectra recorded from synthetic melanin in solution. Excitation and emission conditions are indicated by colors, on the right.

**Figure 2 biology-12-00433-f002:**
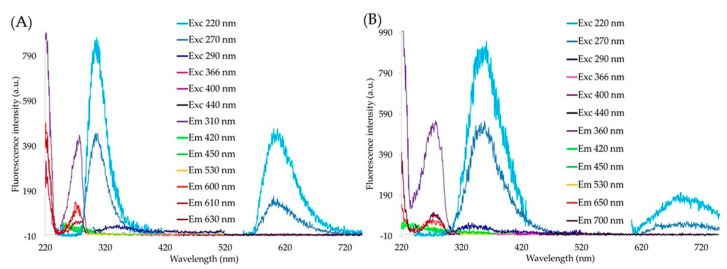
Emission and excitation spectra recorded from L-tyrosine (**A**) and L-tryptophan (**B**) freshly prepared solutions. Excitation and emission conditions are indicated by colors in the inlet legend.

**Figure 3 biology-12-00433-f003:**
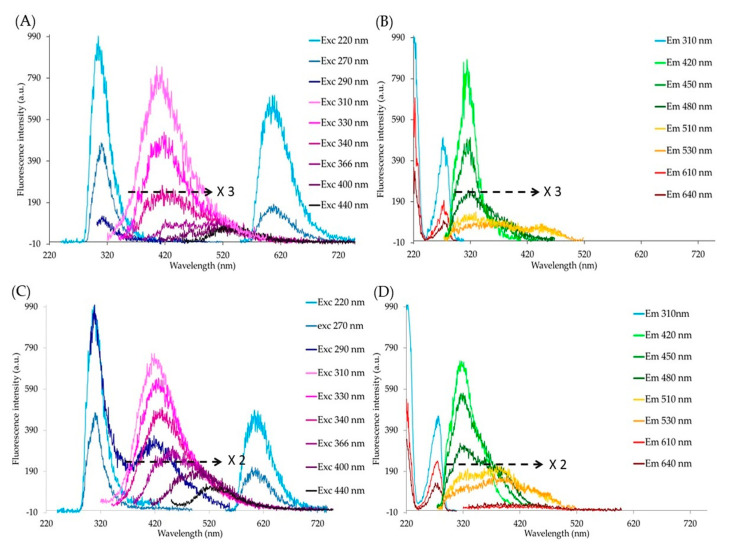
Emission (**A**,**C**) and excitation (**B**,**D**) spectra recorded during the melanization processes, at one (**A**,**B**) and six months (**C**,**D**) of L-tyrosine solution aging. To allow the shapes of the spectra covering the 320–620 nm range (**A**) and the 300–520 nm range (**B**) to be better appreciated, their amplitudes are increased with respect to the fluorescence intensity scale values, as indicated by the arrow in each panel. Excitation and emission conditions are indicated by colors on the right.

**Figure 4 biology-12-00433-f004:**
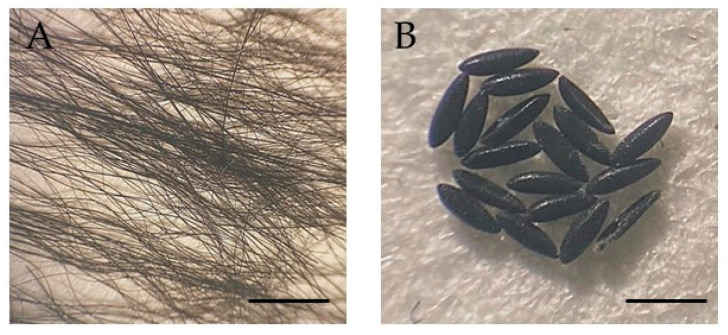
Images of (**A**) black cat hairs and (**B**) mosquito eggs. Bars: 600 μm (**A**), 500 μm (**B**).

**Figure 5 biology-12-00433-f005:**
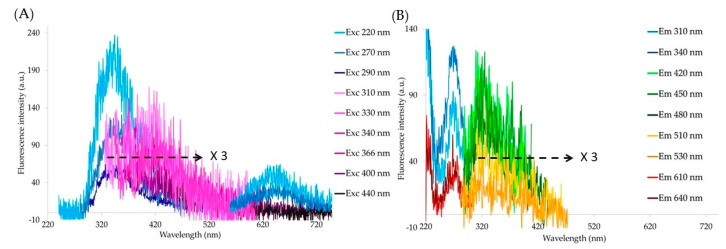
Emission (**A**) and excitation (**B**) spectra recorded from the preparation extract from black cat hairs. To allow the shapes of the spectra covering the 320–620 nm range (**A**) and the 300–470 nm range (**B**) to be better appreciated, their amplitudes are increased with respect to the fluorescence intensity scale a.u. values as indicated by the arrow in each panel. Excitation and emission conditions are indicated by colors on the right.

**Figure 6 biology-12-00433-f006:**
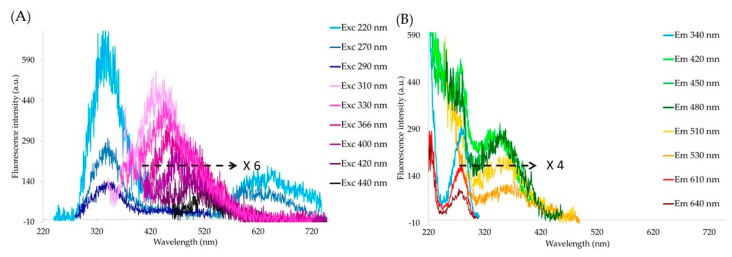
Emission (**A**) and excitation (**B**) spectra recorded from the preparation extract from mosquito eggs. To allow the spectral shapes of the spectra covering the 320–620 nm range (**A**) and the 300–520 nm range (**B**) to be better appreciated, their amplitudes are increased with respect to the fluorescence intensity scale a.u. values as indicated by the arrow in each panel. Excitation and emission conditions are indicated by colors on the right.

## Data Availability

The data presented in this study are available in the article and in the Appendix A file. The raw spectral data are available upon request.

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
