# Peer review of "Characterization of Spontaneous Melanization by Fluorescence Spectroscopy: A Basis for Analytical Application to Biological Substrates"

_biology, 2023, doi:10.3390/biology12030433_

Round 1

Reviewer 1 Report

The authors reported the fluorescence spectra of melanin, L-tryptophan and L-tyrosine at different conditions of measurement. The manuscript is well-structured with proper number of figures. However, there are several aspects that need significant improvements. Before acceptance for publication in Biology, the following revisions should be addressed.

1.      What was the purity of the samples from black cat hairs and mosquito eggs? It should be characterized by some technologies.

2.      There are some grammar errors especially in the section of Results. Some sentences are difficult to understand. The authors should have the manuscript proofread by some experienced researchers.

3.      The sentences in the section of Discussion were too cumbersome and should be simplified.

4.      As described in the manuscript, the spectra of L-tyrosine and melanin were previously reported (References 30, 31, 33, 64, 27-29). So, what is the innovation of this article?

Author Response

Reviewer 1

The authors reported the fluorescence spectra of melanin, L-tryptophan and L-tyrosine at different conditions of measurement. The manuscript is well-structured with proper number of figures. However, there are several aspects that need significant improvements. Before acceptance for publication in Biology, the following revisions should be addressed.

Q 1.      What was the purity of the samples from black cat hairs and mosquito eggs? It should be characterized by some technologies.

A 1. The purity from external contaminants was ensured by the preliminary accurate rinsing of the samples with ultrapure water (a sentence has been added, lines 169-171 of the revised version).

To perform the fluorescence characterization of the extracts, the biological samples were purposely submitted to the mild extraction procedure (as already stated in the previous text, sentences are now at lines 416-425) in the revised version. This was intended to preserve at the best the chemical and photophysical properties of the fluorescing biomolecules.

The attribution of melanin and its precursors, or to L-tyrosine or L-tryptophan to the fluorescing components relied on our work on pure compound aimed to assess their spectral fingerprints, and on the available literature.

Besides, we are aware of the importance of the combination of our mild procedure with additional aggressive treatments and analytical methods, to achieve a comprehensive chemical characterization of melanin related compounds. This is an insight for the future development of the study.

Q 2. There are some grammar errors especially in the section of Results. Some sentences are difficult to understand. The authors should have the manuscript proofread by some experienced researchers.

A2. The text has been carefully checked and revised

Q 3.   The sentences in the section of Discussion were too cumbersome and should be simplified.

A 3.  The sentences have been revised.

Q 4.      As described in the manuscript, the spectra of L-tyrosine and melanin were previously reported (References 30, 31, 33, 64, 27-29). So, what is the innovation of this article?

A 4. We thank the Reviewer for this remark. As from the literature recalled in our text, we are aware and agree that there is no novelty in presenting the spectra of L-tyrosine and melanin. Our intention anyway was to show them as a reference at the convenience of the reader, to make easier to appreciate the changes in the spectra collected during melaninization and from biological samples. This was already suggested in previous text (sentence now at lines 150-154 of the revised version), and stated more specifically by adding new sentences (lines 144-147; lines 229-230 of the revised version).

Reviewer 2 Report

See the attached document

Author Response

Reviewer 2

OBSERVATIONS

regarding the manuscript Fluorescence spectroscopy characterization of spontaneous melaninization as a basis for analytical applications to biological substrates’’

In the paper Fluorescence spectroscopy characterization of spontaneous melaninization as a basis for analytical applications to biological substrates, authors have investigated the fluorescent behaviour of melanin and its precursors.

However, major improvements are needed before being published in the Biology journal:

Q 1. Please check the title of the article. The phrasing is incorrect.

A 1. The title of the article has been revised.

Q 2. The authors say they investigated the emission bands of amino acid solutions below 220 nm The figures in the supplementary material have on-axis values above 220 nm. Please review this.

A 2. Thanks for signaling the misunderstanding, the sentence has been revised.

Q 3. The fluorescence spectra shown are very noisy. Spectral accumulation could have been done to remove some of the noise. In addition, the authors need to remove the inner filter effect, otherwise the interpretation of the results is not correct

A 3. We agree with the Reviewer that spectral accumulation by repeating more times the measurement, can lead to reduction of the noise. However, we had to repeat many measurements on each sample to collect the series of excitation and emission spectra. For this reason, each sample was exposed several times to excitation at various wavelengths. Therefore, we preferred to minimize overexposure by collecting one spectrum for each of the several measurement conditions applied to the same sample.

As to the inner effect, absorption was checked and proper dilutions were performed at the convenience of the linearity of the fluorescence response. New related text has been added at lines 192-195 of the revised version.

Q 4. Usually excitation is done at the wavelength at which the absorbance is maximum (or at a wavelength close to it). The choice to focus on such varied wavelengths must be justified. What is the intended purpose?

A 4. We fully agree that usually in exploring the spectral properties of a fluorescing molecule, excitation to collect the emission spectrum is applied at the wavelength at which the absorbance is maximum.

Anyway, as recalled in our text , for example see the first paragraph of Discussion, in the case of melanin and of its intermediates detectable during melaninization, many different fluorescing components can contribute to at the same time to the overall fluorescence emission. Therefore, we changed excitation and emission conditions to explore the simultaneous presence of various fluorescing components accounting for this behavior.

Q 5. Fluorescence experiments should be complemented by UV-Vis absorption spectroscopy studies. Complementary results (if obtained) could better explain the importance and necessity of the present study.

A 5. We fully agree with the Reviewer on the importance of presenting also UV-Vis absorption spectroscopy data, and sincerely thank for this remark. This helps us to better highlight the usefulness of fluorescence analysis presented in this study, providing insights for next investigations on biological substrates.

A new figure showing absorption spectra has been added (Supplementary section, Figure S1), as well as a sentence explaining the greater ability of fluorescence to provide information on the fluorescing components as compared with absorption (lines 276-281).

Round 2

Reviewer 1 Report

It can be ccepted in present form.

Reviewer 2 Report

No comment